# Examination of the Effect of Task Complexity and Coping Capacity on Driving Risk: A Cross-Country and Transportation Mode Comparative Study

**DOI:** 10.3390/s23249663

**Published:** 2023-12-07

**Authors:** Stella Roussou, Thodoris Garefalakis, Eva Michelaraki, Christos Katrakazas, Muhammad Adnan, Muhammad Wisal Khattak, Tom Brijs, George Yannis

**Affiliations:** 1Department of Transportation Planning and Engineering, National Technical University of Athens, 5 Iroon Polytechniou Str., 157 73 Athens, Greece; tgarefalakis@mail.ntua.gr (T.G.); evamich@mail.ntua.gr (E.M.); ckatrakazas@mail.ntua.gr (C.K.); geyannis@central.ntua.gr (G.Y.); 2Transportation Research Institute (IMOB), School of Transportation Sciences, Hasselt University, 3500 Hasselt, Belgium; muhammad.adnan@uhasselt.be (M.A.); muhammadwisal.khattak@uhasselt.be (M.W.K.); tom.brijs@uhasselt.be (T.B.)

**Keywords:** task complexity, coping capacity, crash risk, generalized linear models, structural equation models

## Abstract

The i-DREAMS project established a ‘Safety Tolerance Zone (STZ)’ to maintain operators within safe boundaries through real-time and post-trip interventions, based on the crucial role of the human element in driving behavior. This paper aims to model the inter-relationship among driving task complexity, operator and vehicle coping capacity, and crash risk. Towards that aim, data from 80 drivers, who participated in a naturalistic driving experiment carried out in three countries (i.e., Belgium, Germany, and Portugal), resulting in a dataset of approximately 19,000 trips were collected and analyzed. The exploratory analysis included the development of Generalized Linear Models (GLMs) and the choice of the most appropriate variables associated with the latent variables “task complexity” and “coping capacity” that are to be estimated from the various indicators. In addition, Structural Equation Models (SEMs) were used to explore how the model variables were interrelated, allowing for both direct and indirect relationships to be modeled. Comparisons on the performance of such models, as well as a discussion on behaviors and driving patterns across different countries and transport modes, were also provided. The findings revealed a positive relationship between task complexity and coping capacity, indicating that as the difficulty of the driving task increased, the driver’s coping capacity increased accordingly, (i.e., higher ability to manage and adapt to the challenges posed by more complex tasks). The integrated treatment of task complexity, coping capacity, and risk can improve the behavior and safety of all travelers, through the unobtrusive and seamless monitoring of behavior. Thus, authorities should utilize a data system oriented towards collecting key driving insights on population level to plan mobility and safety interventions, develop incentives for road users, optimize enforcement, and enhance community building for safe traveling.

## 1. Introduction

Road safety is a pressing global concern as road crashes tragically claim the lives of approximately 1.3 million people each year and result in countless injuries [1]. Factors such as human behavior, road design, vehicle safety features, environmental conditions, and socioeconomic disparities significantly influence the occurrence and severity of road crashes [1]. A substantial portion of these crashes can be attributed to driving behavior, making drivers a vital area of focus in traffic safety research [2]. Recognizing the significance of this issue, the European Union and the World Health Organization have set ambitious targets to reduce fatal traffic crashes by 50% from 2021 to 2030, with emerging technology playing a pivotal role in achieving road safety improvements [3].

A multitude of risk factors influence road safety, including the driver’s state, environmental conditions, and traffic circumstances [4]. Despite advancements in technology and infrastructure, human error remains a significant contributor to traffic collisions [5]. However, the ongoing progress in autonomous vehicles holds promise for enhancing road safety by reducing reliance on human drivers [6]. Additionally, intelligent driving behavior monitoring systems, equipped with real-time interventions, have shown remarkable effectiveness in enhancing road safety [7]. By combining the benefits of autonomous vehicles and intelligent monitoring systems, there is a strong potential for mitigating the impact of human error and creating a safer road environment for all road users.

Numerous studies have focused on understanding the impact of various factors on unsafe driving and have sought to develop suitable models for identifying risky driving behavior and establishing intervention frameworks within vehicles. While there have been proposals for various interventions during and post-trip [8,9], the personalization of these interventions and a direct connection between real-time driving behavior and intervention activation remain areas for improvement.

The i-DREAMS project, funded by the European Commission Horizon 2020 initiative (https://idreamsproject.eu/, accessed on 5 December 2023), aims to address these challenges by establishing, developing, testing, and validating a ‘Safety Tolerance Zone’ (STZ) to ensure safe driving behavior. By continuously monitoring risk factors associated with task complexity (e.g., traffic conditions and weather) and coping capacity (e.g., driver’s mental state, driving behavior, and vehicle status), i-DREAMS aims to determine the appropriate level within the STZ and implement interventions to maintain drivers’ operations within acceptable safety limits. The STZ comprises of three levels: ‘Normal’, ‘Dangerous’, and ‘Avoidable Accident’. The ‘Normal’ level indicates a low likelihood of a crash, while the ‘Dangerous’ level suggests an increased possibility of a crash without inevitability. The ‘Avoidable Accident’ level signifies a high probability of a crash, but it also allows sufficient time for drivers to take action and prevent it. The distinction between the ‘Dangerous’ and ‘Avoidable Accident’ levels lies in the more urgent need for intervention in the ‘Avoidable Accident’ level.

In line with the primary objective of the i-DREAMS project, this study aims to explore the dynamic interplay between task complexity and coping capacity, encompassing both vehicle state and operator state factors. For that purpose, data collected from a naturalistic driving experiment with a sample of 80 drivers were utilized and data from Belgian truck drivers, German drivers, and Portuguese bus drivers were collected and analyzed. For the current study, certain risk explanatory factors and the most reliable indications were evaluated, such as time, distance traveled, headway, speed, forward collision, time of day (lighting indicators), or weather conditions. SEM and GLM analyses were implemented.

The paper is structured in the following manner. At the beginning, a detailed introduction to the project and its general objective is highlighted with a literature review presented concerning the analysis of driving behavior utilizing statistical methods. The research methodology is outlined, including an explanation of collecting the data and the theoretical foundations of the underlying models employed. Finally, the results of the study are presented, followed by significant conclusions regarding the association between key factors like task complexity and coping capacity on risk.

## 2. Literature Review

The ongoing progress in autonomous vehicles, coupled with the implementation of intelligent driving behavior monitoring systems, presents a transformative opportunity for road safety. Autonomous vehicles leverage cutting-edge sensors and artificial intelligence, enabling them to process vast amounts of data swiftly and make split-second decisions, thus reducing the likelihood of crashes caused by human factors. Moreover, intelligent monitoring systems, with their ability to actively intervene in real-time, further enhance road safety by curbing risky behaviors. Moreover, intelligent monitoring systems, with their ability to actively intervene in real time, further enhance road safety by curbing risky behaviors [10].

Recent research also suggests that enhancing driver mindfulness could play a crucial role in improving road safety. Professional drivers, who often face high levels of stress and impulsivity, can benefit from mindfulness interventions tailored to their specific needs [11]. These interventions can help drivers develop self-awareness, improve their ability to focus on the present moment, and make more deliberate choices, ultimately reducing impulsive actions and stress perception. Integrating mindfulness practices with advancements in autonomous vehicles and intelligent monitoring systems could offer a holistic approach to enhancing road safety, addressing both technological and human factors simultaneously [11].

An interesting work conducted by Vrábel et al. [12] examined the relationship between alcohol and driver’s reaction time by analyzing video from two synchronized cameras. Results indicated that driver’s reaction time had a direct impact on the stopping distance. Similarly, Hudec et al. [13] investigated the interconnection between fatal traffic crashes and the driving schools being completed by these drivers. With the key findings revealed through the identification, recognition, and analysis of these traffic crashes, as well as proper actions, it is possible to make some progress towards eliminating their impact.

In addition, another study [14] analyzed traffic crashes caused by technical failure of a vehicle on the basis of long-term data. It was revealed that this cause constituted a low proportion of the total number of road traffic crashes. However, in order to eliminate these traffic crashes and their consequences, it would be beneficial to maintain the best possible technical condition of the vehicles and to contribute to road safety. Moreover, Makka et al. [15] assessed the risks associated with the transport of dangerous flammable substances. The results indicated that mobile resources represented a significant source of risk through the transport of dangerous goods in the event of an emergency occurrence associated with their leakage.

The integration of Structural Equation Modeling (SEM) and Generalized Linear Models (GLMs) provides a comprehensive framework to delve into the intricate dynamics between task complexity, coping capacity, and driving risk. The use of both methodologies allows for a multifaceted exploration of the factors influencing road safety outcomes, contributing to the development of evidence-based interventions for safer driving practices.

Studies have utilized SEM to incorporate crash severity-related features, as well as the number of vehicles involved, into a latent construct termed “crash size” to have a comprehensive and more reliable measure of actual crash severity. SEM is a multivariate statistical method that integrates factor analysis and path analysis. It concretizes latent variables that are difficult to directly observe through several observed variables and establishes the relationship among those latent variables [16]. Moreover, the crash contributing factors have been integrated into unmeasurable variables, namely, driver, road, and vehicle, which, in turn, allows researchers to compare the relative contribution of each latent variable in the crash size [17].

SEM is adept at unraveling complex interconnections among multiple latent variables, providing a comprehensive view of how these factors jointly influence crash risk. SEM’s capability of modeling indirect effects has attracted the attention of naturalistic driving studies [18] and real-time crash prediction models [19] to account for the complex associations between inter-correlated variables. In the context of this research, SEM serves as a powerful tool to explore the multifaceted interactions among vehicle attributes, operator characteristics, and contextual factors, all of which collectively shape the dynamics of risk under varying conditions. Notably, SEM facilitates the integration of operator characteristics, vehicle states, and environmental aspects, thereby fostering a deeper comprehension of their combined influence on driving behavior and crash severity.

The application of Generalized Linear Models (GLMs) stands as a pivotal asset in comprehending the intricate interplay between task complexity, coping capacity, and driving risk [20]. GLMs offer a specialized analytical toolset tailored to the multifaceted challenges posed by road safety data. These datasets often deviate from the norm with non-normal distributions and intricate patterns, presenting a significant analytical hurdle. GLMs, however, adeptly navigate this complexity. Firstly, they demonstrate remarkable adaptability, accommodating the skewed distribution of variables commonly encountered in road safety research, such as crash frequencies, while ensuring the robustness and accuracy of analyses remain intact. Secondly, GLMs effectively model categorical outcomes [21], a frequent occurrence in this field, enabling a deeper exploration of how task complexity and coping capacity influence various levels of risk. Furthermore, GLMs address the issue of heteroscedasticity often observed in road safety data, where the variance of driving risk may vary across different levels of task complexity and coping capacity, ensuring that relationships between these variables and driving risk are precisely captured [20,21]. Lastly, GLMs offer the crucial capability of modeling nonlinear relationships, allowing researchers to unravel complex, nonlinear associations that are frequently found in real-world road safety scenarios but might elude linear models.

By incorporating GLMs into the research methodology, a powerful and tailored approach to examine the intricate relationships between task complexity, coping capacity, and driving risk is gained. GLMs are well-suited to handling the heterogeneity present in road safety data due to variations in road types, regions, and driving conditions. Their adaptability to the specific challenges of road safety data ensures the analyses remain robust and accurate. Ultimately, this integration facilitates a more comprehensive and nuanced understanding of how these factors interrelate, paving the way for the development of targeted interventions aimed at enhancing road safety and effectively reducing the risk of crashes.

The integration of Structural Equation Modeling (SEM) and Generalized Linear Models (GLM) in road safety research offers an intricate yet highly effective framework for understanding the multifaceted dynamics between task complexity, coping capacity, and driving risk. SEM empowers researchers to explore latent variables and complex relationships, aiding in the identification of critical factors that contribute to crash severity. It also excels in modeling indirect effects and examining causal relationships, making it invaluable for studies involving inter-correlated variables. On the other hand, GLMs prove indispensable in tackling the non-normality, heteroscedasticity, and non-linear patterns often found in road safety data, ensuring the accuracy and robustness of analyses. By incorporating these methodologies, research not only gains a deeper comprehension of road safety dynamics but also paves the way for evidence-based interventions that can foster safer driving practices and mitigate the risk of crashes. The synergistic application of SEM and GLM represents a powerful approach to advancing road safety research and enhancing our collective efforts to make roadways safer for all.

## 3. Experiment Description

A naturalistic driving experiment was carried out involving 80 drivers from Belgium (professional truck drivers), Germany (amateur car drivers), and Portugal (professional bus drivers) and a large database of 19,000 trips and 847,711 min was created to investigate the most prominent driving behavior indicators available, including speeding, headway, duration, distance, and harsh events (i.e., harsh acceleration and harsh braking). The total number of drivers, trips, and minutes per country and transport mode is presented in Figure 1 below and a part of the dataset used is located in the Appendix A at the end.

The i-DREAMS on-road experiments were designed utilizing numerous validated concepts adapted from the prior literature, with an emphasis on evaluating interventions that assist drivers to operate within the safety boundaries of the STZ. Driving behavior and the influence of real-time interventions (i.e., in-vehicle warnings) and post-trip interventions (i.e., post-trip feedback and gamification) on driving behavior were thoroughly investigated as part of the on-road experiment. Figure 2 illustrates each of the phases of the i-DREAMS on-road experiment.

Figure 3 demonstrates the most relevant variables utilized for defining task complexity and coping capacity (vehicle and operator state), as well as the variables employed to represent risk. These variables are instrumental to this study, essential for capturing the complex dynamics of the interrelationship between the driving task complexity, operator and vehicle coping capacity, and crash risk.

Regarding weather-related hazards such as rain, snow, or debris, car wipers are considered an important factor in weather conditions. The rate at which the wipers operate can also serve as an indicator of the severity of the weather conditions. For instance, heavy rain or snow can be related to the fast-moving rate of wipers. In contrast, in case the wipers move slowly, it might indicate that there are mild weather conditions. In general, car wipers represent a crucial safety component in a vehicle, aiding drivers in safely maneuvering through diverse weather conditions.

Furthermore, the use of high-beam headlights serves as an indicator of lighting conditions as they are employed to deliver the highest level of illumination during low-light or nighttime driving scenarios. High-beam headlights are specifically engineered to project a more extended beam of light down the road, thereby assisting drivers in identifying potential obstacles or pedestrians that might otherwise be challenging to discern with low-beam headlights. In summary, high-beam headlights constitute a critical vehicle feature that aids drivers in safely maneuvering through varying lighting conditions.

For ‘vehicle state’, three core aspects are considered: technical specifications (including metrics like average speed, braking power, and acceleration performance), actuators and admitted actions (measured through accelerator, brakes, steering wheel usage), and current status (evaluated based on fuel efficiency, scheduled maintenance, real-time data from onboard systems, Telematics/GPS, smartphones, and additional information from ADAS systems such as headway and collision monitoring, pedestrian warning, lane keeping monitoring, onboard cameras, and more).

Regarding ‘operator state’, six key aspects are taken into account: mental state (assessed using metrics on alertness, attention, emotions, etc.), behavior (measured through metrics like speeding, harsh acceleration/braking/cornering, seat belt usage), competencies (evaluated based on metrics concerning risk assessment, attention regulation, self-appraisal, etc.), personality (measured through metrics on adventure-seeking, disinhibition, experience-seeking, boredom susceptibility, etc.), sociodemographic profile (including variables like age, gender, experience, socio-economic status, nationality, ethnicity, cultural identity, etc.), and health status (measured based on metrics regarding current symptoms, neurologic and cardiovascular indicators, medication usage, and other health-related factors). These components collectively contribute to our understanding of coping capacity.

## 4. Equipment Description

In order to collect a range of vehicle and driver-related driving attributes, the project used a system composed of several devices used for collecting data and then implementing real-time interventions. More specifically, data from the Mobileye system (Mobileye, 2022), a dash camera, and the Cardio gateway (CardioID Technologies, 2022), which records driving behavior (e.g., speed, acceleration, deceleration, steering), along with GNSS signals were used. In particular, the Mobileye system is a network sensor and a camera-based system mounted on the windshield that measures parameters like headway monitoring, lane position monitoring, traffic sign recognition, and pedestrian recognition. The system can be connected to the CAN bus and enables integration with several ADAS ecosystem products. The Cardio gateway is a system based on sensors which is connected to the Mobileye equipment through the CAN bus of the vehicle and can transfer data through different communication technologies (BLE, CAN, I2C, SPI, WiFi). Information about the current warning stage, as defined by Mobileye, was also collected for comparison with the i-DREAMS warning stage (i.e., normal driving, danger phase, avoidable accident phase). At the same time, information about the current state of the i-DREAMS platform was collected.

An OBD-II device supporting all OBD-II protocols was installed in each vehicle. A modern vehicle supports hundreds of parameters, which are recorded by the OBD-II device which accommodates the proper Software Development Kit (SDK) to extract the necessary data, as well as a rich set of APIs (Application Programming Interfaces) to communicate with third-party systems. This OBD-II integrates 2G or 3G GSM/GPRS technology through which all data recorded from the vehicle through its sensors are transmitted to remote servers (Cloud). The mobile network is used for data transmission without any user involvement.

CardioID also provided a web API to support data access within the i-DREAMS project. The API completely followed the REST architectural style, and the data were available in a JSON format. For car drivers, Cardio Watch was also used to provide more reliable data about car drivers’ fatigue and sleepiness compared to Cardio Wheel which requires both hands on the steering wheel to provide fatigue index data. In case of trucks and buses, this technology was used as drivers’ does not need to bothered about wearing Cardio Watch which also required timely charging of their batteries.

OSEVEN provided a state-of-the-art android-based smartphone application that also monitors and collects driving behavior of individuals using a variety of parameters. The app uses different smartphone sensors to collect such data. The app was used by drivers recruited for on-field trials. Drivers recruited for the experiments were required to install this app on their smartphone. The app does not collect any data but was used to display processed data so that driving behavior can be influenced, especially for trucks and buses. For cars, the O7 app provided this functionality along with collection of detailed data. Due to differences between modes, their operation, technological capabilities, and vehicle design, the data collection methods and measures differ between the on-road field trials.

## 5. Algorithm and Experiment’s Methodology Description

Raw data for a particular trip were collected via CardioID gateway, Mobileye, wristband, or CardioWheel. These trip data were fused using a feature-based data fusion technique, namely geolocation through synchronization and support vector machines. The system provided by CardioID integrates several data streams, generated by the different sensors that make up the inputs of the i-Dreams system. After data validation, confirming that no unordered or repeated points exist in the data set, events generated by the analysis of accelerometer data and by the Mobileye were geolocated through synchronization with the GPS information. Moreover, time on task was fused with heart rate variability (HRV) as features used by CardioID’s sleepiness machine learning model to produce the sleepiness state output. Data fusion was also implemented on processed data in i-DREAMS. The processed data had already been fused once before processing (i.e., all of the previously mentioned data fusion techniques on raw data).

However, such fused data still needed aggregation for specific data analysis needs in i-DREAMS. The aggregation method depends on the type of data and their purpose for analysis. For example, in the case of event data, the aggregation method used is the counting of events in a specified interval; in the case of continuous data such as speed, headway, etc., the aggregation was performed by calculating their mean, minimum, and maximum in a specific interval. Also, the nature of KSS (being discrete) requires a median as the aggregation method. In addition, trip level data also contain scores information on various risk indicators (which is derived from the event type and their frequency). A number of python scripts were developed to convert the available trip data (collected in the i-Dreams back-office) into its fused form so that the data are ready for analysis.

## 6. Methods

### 6.1. Generalized Linear Models (GLMs)

GLMs represent a versatile extension of standard linear regression. They accommodates response variables with error distribution models that deviate from the normal distribution. GLMs extend linear regression by establishing a connection between the linear model and the response variable through a link function and by permitting the variance of each measurement to be influenced by its predicted value [22].

In a GLM, each outcome Y of the dependent variables is assumed to be generated from a particular distribution in an exponential family, e.g., normal, binomial, Poisson, and gamma distributions, among others. The mean, μ, of the distribution depends on the independent variables, X, through:(1)$$Ε\left(Y|X\right)=\mu =g-1\left(Χ\beta \right),$$
where E(Y|X) is the expected value of Y conditional on X; Xβ is the linear predictor, a linear combination of unknown parameters β; and g is the link function.

In this framework, the variance is typically a function, V, of the mean:(2)$$Var\left(Y|X\right)=V\left(g-1\left(Χ\beta \right)\right),$$

It is convenient if V follows from an exponential family of distributions, but it may simply be that the variance is a function of the predicted value.

The unknown parameters, β, are typically estimated with maximum likelihood, maximum quasi-likelihood, or Bayesian techniques.

GLMs were formulated as a way of unifying various other statistical models, including linear regression, logistic regression, and Poisson regression. In particular, ref. [23] proposed an iteratively reweighted least squares method for maximum likelihood estimation of the model parameters. Maximum-likelihood estimation remains popular and is the default method on many statistical computing packages. Other approaches, including Bayesian approaches and least squares fits to variance-stabilized responses, have been developed.

A key point in the development of the GLM was the generalization of the normal distribution (on which the linear regression model relies) to the exponential family of distributions. This idea was developed by [24]. Consider a single random variable y whose probability (mass) function (if it is discrete) or probability density function (if it is continuous) depends on a single parameter θ. The distribution belongs to the exponential family if it can be written as follows:(3)$$f\left(y;\theta \right)=s\left(y\right)t\left(\theta \right)e\left(a\left(y\right)b\left(\theta \right)\right),$$
where a, b, s, and t are known functions. The symmetry between y and θ becomes more evident if the equation above is rewritten as follows:(4)$$f\left(y;\theta \right)=\exp\left[a\left(y\right)b\left(\theta \right)+c\left(\theta \right)+d\left(y\right)\right],$$
where $s\left(y\right)=exp[d\left(y\right)]$ and $t\left(\theta \right)=exp[c\left(\theta \right)]$.

If a(y) = y then the distribution is said to be in the canonical form. Furthermore, any additional parameters (besides the parameter of interest θ) are regarded as nuisance parameters forming parts of the functions a, b, c, and d, and they are treated as though they are known. Many well-known distributions belong to the exponential family, including Poisson, normal, or binomial distributions. On the other hand, examples of well-known and widely used distributions that cannot be expressed in this form are Student’s t-distribution and the uniform distribution.

It should be mentioned that the Variance Inflation Factor (VIF) is a measure of the amount of multicollinearity in regression analysis. Multicollinearity exists when there is a correlation between multiple independent variables in a multiple regression model. The default VIF cutoff value is 5; only variables with a VIF less than 5 will be included in the model (VIF < 5). However, in certain cases, even if the VIF is less than 10 it can be accepted.

### 6.2. Structural Equation Models (SEMs)

Structural Equation Modelling or path analysis is a multivariate method used to test hypotheses regarding the influences among interacting observed and unobserved variables. The observed variables are measurable, while unobserved variables are latent constructs.

Structural equation models consist of two components: a measurement model and a structural model. The measurement model is used to assess how well various observable exogenous variables can measure the latent variables, as well as the measurement errors associated with them. The structural model is used to investigate the relationships among the model variables, enabling the modeling of both direct and indirect linkages. In this regard, SEMs distinguish themselves from regular regression techniques by deviating from direct relationships between variables.

The general formulation of an SEM is as follows [25,26]:(5)η=βη+γξ+ε,

In Equation (5), η represents a vector of endogenous variables, ξ represents a vector of exogenous variables, β and γ are vectors of coefficients to be estimated, and ε represents a vector of regression errors.

The measurement models can be described as follows [27]:(6)x=Λxξ+δ, for the exogenous variables
(7)y=Λyη+ζ, for the endogenous variables

In Equations (6) and (7), x and δ represent vectors associated with the observed exogenous variables and their errors, while y and ζ are vectors that represent vectors associated with the observed endogenous variables and their errors. Λx and Λy are structural coefficient matrices that capture the effects of the latent exogenous and endogenous variables on the observed variables.

To depict the structural model, path analysis is often employed, illustrating how a set of “explanatory” variables can influence a “dependent” variable. The paths can be visually represented to indicate whether the explanatory variables are correlated causes, mediated causes, or independent causes of the dependent variable.

### 6.3. Model Goodness-of-Fit Measures

In the context of model selection, model Goodness-of-Fit measures comprise an important part of any statistical model assessment. Several goodness-of-fit metrics are commonly used, including the goodness-of-fit index (GFI), the (standardized) Root Mean Square Error Approximation (RMSEA), the comparative fit index (CFI), and the Tucker–Lewis Index (TLI). Such criteria are based on differences between the observed and modeled variance–covariance matrices. A detailed description of the aforementioned metrics is presented below.

The Akaike Information Criterion (AIC), which accounts for the number of included independent variables, is used for the process of model selection between models with different combinations of explanatory variables [28].
(8)$$AIC=-2L\left(\theta \right)+q,$$
where q is the number of parameters and L(θ) is the log-likelihood at convergence. Lower values of AIC are preferred to higher values because higher values of −2L(θ) correspond to a greater lack of fit.

The Bayesian Information Criterion (BIC) is used for model selection among a finite set of models; models with a lower BIC are generally preferred.
(9)$$BIC=-2L\left(\theta \right)+qln\left(N\right),$$

The Akaike Information Criterion (AIC) and the Bayesian Information Criterion (BIC) provide measures of model performance that account for model complexity. AIC and BIC combine a term reflecting how well the model fits the data with a term that penalizes the model in proportion to its number of parameters.

The Comparative Fit Index (CFI) is based on a noncentral x^2^ distribution. It evaluates the model fit by comparing the fit of a hypothesized model with that of an independence model. The values of the CFI range from 0 to 1, indicating a good fit for the model when the value exceeds 0.95 [29]. In general, values more than 0.90 for the CFI are generally accepted as indications of very good overall model fit (CFI > 0.90). The formula is represented as follows:(10)$$CFI=1-\frac{max({x_{H}}^{2}-{df}_{H},0)}{max({x_{H}}^{2}-{df}_{H},{x_{I}}^{2}-{df}_{I})},$$
where x^2^_H_ is the value of x^2^ and df_H_ is the degrees of freedom in the hypothesized model, and x^2^_I_ is the value of x^2^ and df_I_ is the degrees of freedom in the independence model.

The Tucker–Lewis Index (TLI) considers the parsimony of the model. Therefore, if the fit indices of the two models are similar, a simpler model (i.e., greater degrees of freedom) is chosen. TLISI is an unstandardized value, so it can have a value less than 0 or greater than 1. It indicates a good fit for the model when the value exceeds 0.95 [29]. In general, values more than 0.90 for the TLI are generally accepted as indications of a very good overall model fit (TLI > 0.90). The formula is represented as follows:(11)$$TLI=\frac{\frac{{x_{I}}^{2}}{{df}_{I}}-\frac{{x_{H}}^{2}}{{df}_{H}}}{\frac{{x_{I}}^{2}}{{df}_{I}}-1},$$
where x^2^_H_ is the value of x^2^ and df_H_ is the degrees of freedom in the hypothesized model, and x^2^_I_ is the value of x^2^ and df_I_ is the degrees of freedom in the independence model.

Currently, one of the most widely used goodness-of-fit indices is the Root Mean Square Error Approximation (RMSEA). The RMSEA measures the unstandardized discrepancy between the population and the fitted model, adjusted by its degrees of freedom (df). Different proposals have been made as to the correct use of the RMSEA. The most common approach is to calculate and interpret the sample’s RMSEA [30]. The RMSEA is considered a “badness-of-fit measure”, meaning that lower index values represent a better-fitting model. The RMSEA index ranges between 0 and 1. Its value 0.05 or lower is indicative of a model fit with observed data. p close value tests the null hypothesis that the RMSEA is no greater than 0.05. If the p close value is more than 0.05, the null hypothesis is accepted that the RMSEA is no greater than 0.05 and indicates the model closely fits the observed data (RMSEA < 0.05). The formula is represented as follows:(12)$$RMSEA=\sqrt{\frac{{x_{H}}^{2}-{df}_{H}}{{df}_{H}(n-1)}},$$
where x^2^_H_ is the value of x^2^ and df_H_ is the degrees of freedom in the hypothesized model; n is the sample size.

The Root Mean Squared Error (RMSE) is one of the most commonly used measures for evaluating the quality of predictions. It shows how far predictions fall from measured true values using Euclidean distance.

The formula of the RMSE, which is the square root of the average squared error, is represented as follows:(13)$$RMSE=\sqrt{\frac{1}{N}\sum e_{t}^{2}}$$
where N is the number of forecasted points and et is the error (i.e., observedt—forecastedt).

The Goodness of Fit Index (GFI) is a measure of fit between the hypothesized model and the observed covariance matrix. The adjusted Goodness of Fit Index (AGFI) corrects the GFI, which is affected by the number of indicators of each latent variable [31]. The GFI and AGFI range between 0 and 1, with a value of over 0.9 generally indicating an acceptable model fit. In general, values more than 0.90 for GFI are generally accepted as indications of a very good overall model fit (GFI > 0.90).

Lastly, the Hoelter index is calculated to find if the chi-square is insignificant or not. If its value is more than 200 for the model, then the model is considered to be a good fit with observed data (Hoelter > 200). Values of less than 75 indicate a very poor model fit. The Hoelter only makes sense to interpret if N > 200 and the chi-square is statistically significant.

## 7. Results

### 7.1. GLM Results

Generalized Linear Models (GLMs) were utilized to examine the connection between a crucial speed-related performance metric for truck drivers in Belgium, car drivers in Germany, and bus drivers in Portugal. In order to implement the GLM analyses it must be noted that the data used were from all four phases of each country experiment. The link between speeding and risk is well-established within the road safety field, and consequently, speeding frequently serves as a standard dependent variable in research related to human factors in transportation.

#### 7.1.1. Belgian Trucks

The first GLM investigated the relationship between speeding and several explanatory variables of task complexity and coping capacity (operator state) in Belgium. Specifically, the developed model employs the dependent variable “speeding”, represented by a binary variable: coded as 1 to indicate the presence of a speeding event and as 0 if there is no such event. Regarding task complexity, the model incorporates variables such as time indicators, wipers, and high beam. Meanwhile, for assessing coping capacity (operator state), the model considers variables including distance traveled and instances of harsh acceleration. It should be noted that certain explanatory variables related to vehicle state, such as fuel type, vehicle age, or gearbox type, and socio-demographic characteristics like gender, age, or educational level, were found to lack statistical significance at a 95% confidence level. Consequently, these variables have been excluded from the models. The model parameter estimates are presented in Table 1.

It should be noted that the intercept variable signifies the expected value of the dependent variable when all independent variables are set to zero. In particular, it represents the baseline or starting point of the relationship between the dependent and independent variables.

According to the data presented in Table 1, it becomes evident that all explanatory variables exhibit statistical significance at a confidence level of 95%. Furthermore, there are no concerns about multicollinearity as the Variance Inflation Factor (VIF) values are significantly below the threshold of 5. Regarding the coefficient analysis, it was uncovered that indicators associated with task complexity, such as the time indicator and the use of wipers, displayed positive correlations with speeding. The time indicator, in particular, signifies different times of the day (day represented as 1, dusk as 2, and night as 3), indicating that higher speeding events occur at night compared to during the day. This might be caused by a smaller number of vehicles on the road, lower visibility, and a misleading sense of security that comes with driving in the dark. It is worth noting that wipers (wipers off coded as 0, wipers on coded as 1) were also found to have a positive correlation with speeding. This suggests that there is a higher occurrence of speeding incidents during adverse weather conditions, such as rainy weather. This phenomenon could be attributed to the challenges posed by wet and slippery roads, making it harder to maintain control of the vehicle. Furthermore, rain can impair visibility, making it more difficult to spot other vehicles or obstacles on the road. Considering the indicator for high beam usage (indicating lighting conditions, with no high beam detected), a negative correlation was identified. This implies that when the high beam was not in use (typically during daytime driving), fewer speeding incidents were recorded. This finding aligns with the earlier observation regarding the time of day, indicating that higher instances of speeding tend to occur at night compared to other times of the day.

Regarding the indicators reflecting coping capacity (operator state), there was a positive association between harsh accelerations and the dependent variable (speeding). This implies that an increase in the number of abrupt accelerations corresponds to an uptick in speeding occurrences. This finding holds significance as it reinforces the statistically significant positive correlation between speeding and instances of aggressive driving behavior. Lastly, the total distance traveled was found to be negatively correlated with speeding. This might be attributed to the notion that the longer a person drives, the more likely they are to experience fatigue, leading them to drive at reduced speeds and exercise greater caution.

#### 7.1.2. German Cars

The second GLM examined how speeding relates to various explanatory factors of task complexity and coping capacity (pertaining to both vehicle and operator state) within the context of Germany. For task complexity, the variables used are time indicator and high beam, for coping capacity (vehicle state), the variables used are a type of fuel and vehicle age, while for coping capacity (operator state), the variables used are distance traveled, duration, harsh acceleration, drowsiness, gender, and age. The model parameter estimates are summarized in Table 2.

Based on Table 2, it can be observed that all explanatory variables are statistically significant at a 95% confidence level; there is no issue of multicollinearity (VIF < 5). With regard to the coefficients, it was revealed that the indicators of task complexity, such as time and high beam (indicating lighting conditions; no high beam detected) were positively correlated with speeding. Regarding the indicators of coping capacity (vehicle state), fuel type and vehicle age were positively correlated with speeding. Furthermore, it was demonstrated that indicators of coping capacity (operator state), such as harsh accelerations, distance, duration, and drowsiness, had a positive relationship with the dependent variable (i.e., speeding), indicating that as the values of the aforementioned independent variables increase, speeding also increases. This is a noteworthy finding of the current research as it confirms that harsh driving behavior events present a statistically significant positive correlation with speeding.

Taking into consideration socio-demographic characteristics, gender and age were negatively correlated with speeding. In particular, the negative value of the “Gender” coefficient implied that as the value of the variable was equal to 1 (males coded as 0, females as 1), the speeding percentage was lower. Results revealed that the vast majority of male drivers displayed less cautious behavior during their trips and exceeded more often the speed limits than female drivers. It is also remarkable that the negative value of the “Age” coefficient implied that as the value of the variable increased (a higher value indicates increased age and, therefore, increased years of participant’s experience), the speeding percentage was lower. Young drivers appeared to have riskier driving behavior than older drivers and were more prone to exceed the speed limits.

#### 7.1.3. Portuguese Buses

The third GLM investigated the relationship between speeding and several explanatory variables of task complexity and coping capacity (vehicle and operator state) in Portugal. More specifically, for task complexity, the variable used is time indicator, while for coping capacity (operator state), the variables used are distance traveled, harsh acceleration, harsh braking, and fatigue. It should be mentioned that the explanatory variables of vehicle state, such as fuel type, vehicle age, or gearbox, or socio-demographic characteristics, such as gender, age, or educational level, are not statistically significant at a 95% confidence level; thus, these variables are not included in the models. The model parameter estimates are summarized in Table 3.

It can be observed that all explanatory variables are statistically significant at a 95% confidence level; there is no issue of multicollinearity (VIF < 5). With regard to the coefficients, it was revealed that the indicators of task complexity, such as time indicator, were positively correlated with speeding. Time indicator refers to the time of the day (day coded as 1, dusk coded as 2, night coded as 3), which means that higher speeding events occur at night compared to during the day. This may be due to fewer cars on the road, lower visibility, and a false sense of security that comes with driving in the dark. Regarding the indicators of coping capacity (operator state), distance, and harsh events (i.e., harsh acceleration and harsh braking) had a positive relationship with the dependent variable (i.e., speeding), indicating that as the total distance traveled and the number of harsh events increases, speeding also increases. Lastly, fatigue was negatively correlated with speeding, which implies that the more fatigued the driver is the slower and more cautiously they drive.

### 7.2. SEM Results

In order to investigate the relationship between the latent variables of task complexity, coping capacity, and risk (represented as the three stages of the STZ), four distinct SEM models were developed.

#### 7.2.1. Belgian Trucks

Figure 4 illustrates the results for each phase. The latent variable risk is measured by the means of the STZ levels for acceleration (level 1 ‘normal driving’ used as the reference case), with negative correlations of risk with the STZ indicators. The negative sign shows that the latent variable risk could in fact be representing an inverse of risk, more like normal driving. The structural model between the latent variables shows some interesting findings: first, task complexity and coping capacity are interrelated with a positive correlation. This positive correlation indicates that higher task complexity is associated with higher coping capacity, implying that drivers’ coping capacity increases as the complexity of the driving task increases.

Task complexity increase is associated with higher (risk) normal driving (lower risk), which is not intuitive. Although the initial assumption was that task complexity would increase risk or decrease normal driving, once its effect is moderated by that of coping capacity the opposite is the case. It is noted, however, that the task complexity latent variable is measured by environmental indicators (i.e., rainy weather) and situational indicators (i.e., speed) which are known to induce compensatory behaviors by drivers, in particular expressed as reduced speed during more demanding conditions.

At the same time, coping capacity is negatively associated with normal driving or inverse of risk, again an interesting finding. It could be assumed that a higher coping capacity might reduce risk or improve normal driving, but this is not the case here. Furthermore, the coping capacity indicators in our sample include static demographic and self-reported behavior indicators and therefore are more representative of driver personality and general driving styles and less so of the real-time operator state during the experiment. For instance, indicators related to the level of sleepiness, fatigue, or distraction were either not available or not significant in this model. Therefore, it can be concluded that younger, more confident truck drivers exhibited (higher risk) lower normal driving in this experiment, in terms of exceeding the STZ acceleration boundaries, without however taking into account the variations in their state during these trips.

It is observed that the relationships among risk, task complexity, and coping capacity are consistent between the different phases (except for phase 3 where coping capacity and risk have a positive relationship). In particular, in phase 3, the structural relationship between coping capacity and (inverse) risk changes to a positive coefficient. This finding may not be directly interpreted, but it is possible that the presence of real-time and post-trip i-DREAMS interventions in phase 3 led to a different interaction between the latent variables coping capacity and risk, which would need additional indicators available to conclude. Also, the magnitude of the correlation between the latent variables coping capacity and task complexity reduces to an extremely small value.

The loading of ‘trip duration’ in phase 2 changes to a positive sign, which shows an improvement in the coping capacity of drivers in the presence of real-time interventions. However, in the later phases 3 and 4, this trend is back to phase 1. The loadings of the observed proportions of the STZ of acceleration are consistent between the different phases (the loadings of the second STZ level have consistently higher negative signs across all phases, while the loadings of the third STZ level have consistently lower signs across all phases). The loading of the first STZ level becomes notably higher in the fourth phase of the experiment. This may indicate that drivers tend to have normal driving in the fourth phase in the presence of all interventions.

Looking at the observed risk factors, it was demonstrated that for harsh accelerations in Belgian trucks, the correlation of coping capacity and task complexity was in general positive along the same magnitude for all phases

#### 7.2.2. German Cars

To begin with, the risk is measured by means of the STZ levels for speeding (level 1 ‘normal driving’ used as the reference case; level 2 refers to ‘dangerous driving’, while no incidents with regard to level 3 ‘avoidable accident driving’ were found). In particular, positive correlations of risk with the STZ indicators were found. It should be noted that the identified model indicated that level 3 of the speeding variable does not have significant loading in the measurement model for the latent variable risk and thus, this level was not included in the final model. Level 1 and level 2 of speeding (or STZ 1 and STZ 2 indicators) have positive loadings in correlation to the latent variable risk, respectively.

The latent variable task complexity is measured by means of the environmental indicator of time of the day. It should be noted that based on the definition of task complexity, road layout, time, location, traffic volumes, and weather variables should be included in the analysis. However, road type (i.e., urban, rural, highway), location, traffic volumes (i.e., high, medium, low), and weather were not available in the German dataset. Thus, only the time indicator was able to be used in the models applied. To that aim, exposure indicators, such as trip duration and distance traveled, were included in the task complexity analysis. In particular, time of the day, distance, and duration were found to have a positive correlation with task complexity.

Furthermore, it is shown that the latent coping capacity is measured by means of both vehicle state indicators, such as age of the vehicle, gearbox (i.e., automatic or manual), and type of fuel (i.e., diesel, hybrid electric, petrol). At the same time, operator state indicators, such as “Gender” (indicating the gender of the driver; male or female) and “Age” (indicating the age of the driver) are included in the SEM applied.

The structural model between the latent variables shows some interesting findings: First, task complexity and coping capacity are interrelated with a positive correlation (regression coefficient = 0.03), which reduces in magnitude as the drivers progress from phases 1 and 2 through phases 3 and 4. This positive correlation indicates that higher task complexity is associated with higher coping capacity, implying that drivers’ coping capacity increases as the complexity of the driving tasks increases. Overall, the structural model between task complexity and risk shows a positive coefficient, which means that increased task complexity relates to increased risk according to the model (regression coefficient = 2.19). On the other hand, the structural model between coping capacity and risk shows a negative coefficient, which means that increased coping capacity relates to decreased risk according to the model (regression coefficient = −0.05).

It is identified that the measurement equations of task complexity and coping capacity are consistent between the different phases. At the same time, the loadings of the observed proportions of the STZ of speeding are consistent between the different phases. The structural model between task complexity and inverse risk (normal driving) are positively correlated among the four phases, while coping capacity and risk were found to have a negative relationship in all phases of the experiment.

In Germany, the model for speeding revealed a positive correlation between task complexity and coping capacity, but with the largest correlation in phase 2 of the experiment, where real-time warnings were introduced. At the end of the experiment (phase 4), coping capacity was found to have its largest correlation with risk, while task complexity had its greatest loading during phase 3 of the experiment.

The results for all phases are shown in Figure 5 below.

#### 7.2.3. Portuguese Buses

With regard to Portuguese buses, negative correlations of risk with the STZ indicators were found.

The latent variable task complexity is measured by means of the environmental indicator of the time of the day and total duration. In addition, exposure indicators, such as trip duration, were included in the task complexity analysis. In particular, time of the day and duration were found to have a positive correlation with task complexity.

Moreover, it is shown that the latent coping capacity is measured by means of operator state indicators, such as average speed, distance, harsh acceleration, and harsh braking. It should be noted that vehicle state indicators, such as vehicle age, gearbox, fuel type, or socio-demographic characteristics were not provided.

The structural model between the latent variables shows some interesting findings: First, task complexity and coping capacity are interrelated with a positive correlation (regression coefficient = 0.96), which reduces in magnitude as the drivers progress from phases 1 and 2 through phases 3 and 4. This positive correlation indicates that higher task complexity is associated with higher coping capacity, implying that drivers’ coping capacity increases as the complexity of the driving tasks increases. Overall, the structural model between task complexity and risk shows a positive coefficient, which means that increased task complexity relates to increased risk according to the model (regression coefficient = 5.36). On the other hand, the structural model between coping capacity and risk shows a negative coefficient, which means that increased coping capacity relates to decreased risk according to the model (regression coefficient = −5.02).

The results for all phases are shown in Figure 6 below. It is observed that the measurement equations of task complexity and coping capacity are consistent between the different phases. The structural model between task complexity and inverse risk (normal driving) is positively correlated in phases 1, 3, and 4, while a negative correlation in phase 2 was identified. At the same time, coping capacity and risk were found to have a negative relationship in all phases of the experiment.

In Portugal, task complexity was positively associated with the latent variable risk, which was defined by different levels of headway. The higher the complexity, the higher the chance to drive normally and more carefully. On the other hand, coping capacity was negatively associated with risk (or normal driving), which implied that higher coping capacity might encourage normal driving and reduce risk. Task complexity and coping capacity were interrelated with a positive correlation, which reduced in magnitude as the drivers progressed from Phase 1 through Phase 4. Similar patterns of professional drivers (in terms of loadings and signs among phases for Belgian truck and Portuguese bus drivers) were observed.

Table 4 summarizes the model fit of the SEM applied for different counties (Germany, Belgium, Portugal), transport modes (cars, trucks, buses), and experimental phases.

## 8. Discussion

With the escalation of task complexity, drivers could encounter heightened cognitive burden and fragmented focus, which could potentially result in diminished situational awareness and delayed reaction times. These elements have the potential to compromise the capacity for effective decision making and elevate the chances of errors or collisions.

The findings revealed that increasing task complexity was related to an increased crash probability due to a variety of causes. For instance, drivers may feel overwhelmed by the demands of complex tasks, causing them to focus less on the road and other traffic participants. This may lead to a delayed identification of crucial incidents and insufficient reactions. Furthermore, intricate tasks could necessitate drivers to allocate additional mental resources, potentially causing them to shift their focus away from crucial driving tasks. For instance, interacting with in-vehicle technology or navigation systems can increase cognitive workload and lead to decreased focus on the primary task of driving.

The results of this research align with the study of Onate-Vega et al. 2020 [32], in which visual-manual distraction results in a higher standard deviation of speed, indicating increased speed variability. Additionally, Wang et al. 2022 [33] agrees with the results of this research as it is mentioned that highly complex secondary tasks overwhelm the driver’s capacity, leading to crashes.

Conversely, an increased risk of crashes is observed in situations where a driver’s coping capacity is insufficient to effectively manage complex tasks. This reduced coping capacity is evident through slower reaction times, impaired judgment, and difficulties in prioritizing information. Such limitations can lead to errors, misjudgments, and collisions when the demands of the driving task exceed the driver’s coping capacity.

It is important to highlight that the association between task complexity and risk, as well as coping capacity and risk, could be influenced by the context and the nature of the task or activity involved. In general, higher task complexity may increase the potential for errors or crashes as it can lead to greater cognitive or physical demands on the individual performing the task. Nonetheless, it is worth noting that heightened experience or training have the potential to alleviate the risks linked to greater task complexity. Likewise, an enhanced coping capacity can contribute to a decreased risk of crashes or mistakes as it furnishes individuals with the necessary resources or strategies for effectively navigating demanding or stressful situations. However, the efficiency of these coping strategies may hinge on the particular circumstances and an individual’s capability to employ them in practical, real-world scenarios. In summary, it is crucial to take into consideration the unique factors and context when evaluating the interplay among task complexity, coping capacity, and risk.

The models developed have the potential to be utilized more extensively by both researchers and professionals. For instance, researchers and practitioners could incorporate additional variables such as road type, a broader range of personality traits, and diverse driving profiles. Moreover, there is room for improvement in the data used for these models. Enhancements could involve incorporating additional measurements such as readings from electrocardiograms and electroencephalograms, as well as data related to traffic conflicts and transportation emissions. Lastly, it would be beneficial to explore supplementary methodologies, such as imbalanced learning and models that account for unobserved differences among individuals. These approaches could provide valuable insights into understanding how task complexity, coping capacity, and crash risk are interconnected.

In summary, the inter-relationship between driving task complexity, coping capacity, and crash risk is a versatile and critical area of study in traffic safety research. Driving task complexity refers to the level of demand and cognitive strain placed on drivers due to various factors like traffic volume, road conditions, weather, and the presence of distractions. In contrast, coping capacity encompasses an individual driver’s capacity to skillfully manage and adapt to these intricate driving tasks. This encompasses aspects such as driver experience, skills, perceptual abilities, decision-making processes, and the availability of suitable coping strategies. The interaction between driving task complexity and coping capacity had a direct impact on crash risk. Drivers who find themselves overwhelmed by high task complexity and possess limited coping capacity may encounter reduced situational awareness, delayed reaction times, compromised decision-making abilities, and an elevated likelihood of making mistakes or being involved in collisions. On the other hand, drivers with better coping capacity can effectively handle complex driving tasks, mitigate risks, and maintain safer driving behaviors.

Future research could also examine the usefulness of deep learning (DL) techniques on this matter, such as Long Short-Term Memory (LSTM) [34,35]. DL models are increasingly utilized due to their ability to capture complex temporal dependencies of features, thus potentially improving the accuracy and predictive capabilities of driver behavior classification models. Furthermore, the examination of additional data sources, as well as the comparison of naturalistic driving and simulator experiment datasets involving drivers from different countries or transport modes, would assist in the comprehensive understanding and evaluation of the models utilized.

## 9. Conclusions

The ultimate goal of the analyses in this work was to identify the impact that the balance between task complexity and coping capacity has on the risk of a crash. To that end, 80 drivers participated in a naturalistic driving experiment carried out in three countries (i.e., Belgium, Germany, and Portugal) and a large dataset of 19,000 trips was collected and analyzed. Explanatory variables of risk and the most reliable indicators, such as time headway, distance traveled, speed, forward collision, time of the day (lighting indicators), or weather conditions, were assessed.

To fulfill the aforementioned objective, exploratory analyses, such as GLMs, were developed, and the most appropriate variables associated with the latent variables “task complexity” and “coping capacity” were estimated. Moreover, SEMs were also used to explore how the model variables were interrelated.

The results demonstrated that as task complexity levels increased, so did coping capacity. This suggests that drivers tend to effectively manage their ability to anticipate and handle potential challenges when faced with demanding conditions while driving. It was observed that task complexity and inverse risk exhibited a positive correlation in all phases of the experiment, signifying that greater task complexity corresponded to increased risk. Conversely, coping capacity and inverse risk displayed a negative relationship across all phases, indicating that higher coping capacity was associated with reduced risk. Overall, the interventions had a positive impact on risk reduction by enhancing the coping capacity of the drivers.

The integrated treatment of task complexity, coping capacity, and risk can improve the behavior and safety of all travelers through the unobtrusive and seamless monitoring of behavior. Thus, authorities may use data systems at the population level to plan mobility and safety interventions, set up road user incentives, optimize enforcement, and enhance community building on safe traveling.

In order to develop targeted interventions and countermeasures to enhance traffic safety and reduce crash risk on roadways, it is important to investigate and model the aforementioned inter-relationship between task complexity, coping capacity, and crash risk. This involves improving road infrastructure, establishing appropriate signs and road markings, providing education to the drivers about the impact of task complexity on their performance, and promoting the development of coping strategies to manage complex driving situations. Lastly, technological advancements in vehicle automation and driver assistance systems can play a role in mitigating crash risk by reducing the cognitive load associated with complex tasks and providing support to drivers in challenging driving conditions.

## Figures and Tables

**Figure 1 sensors-23-09663-f001:**
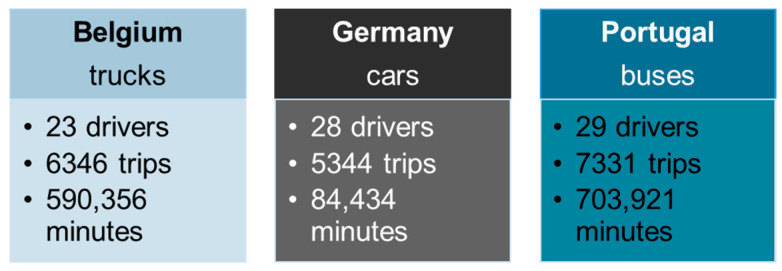
Number of drivers, trips, and minutes per country and transport mode.

**Figure 2 sensors-23-09663-f002:**
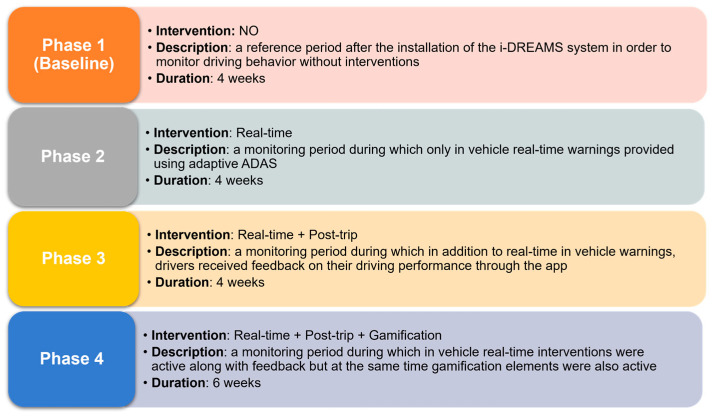
Overview of the different phases of the experimental design.

**Figure 3 sensors-23-09663-f003:**
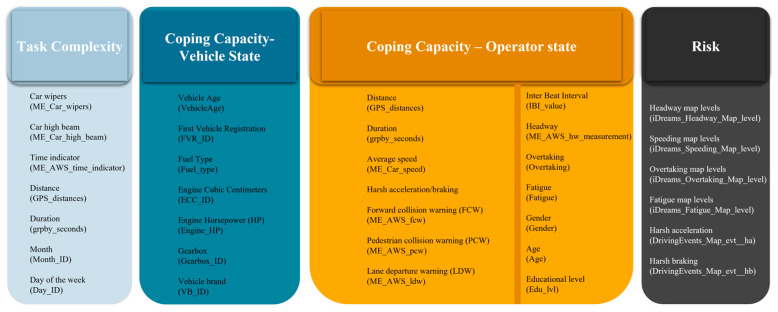
Variables for task complexity and coping capacity and risk.

**Figure 4 sensors-23-09663-f004:**
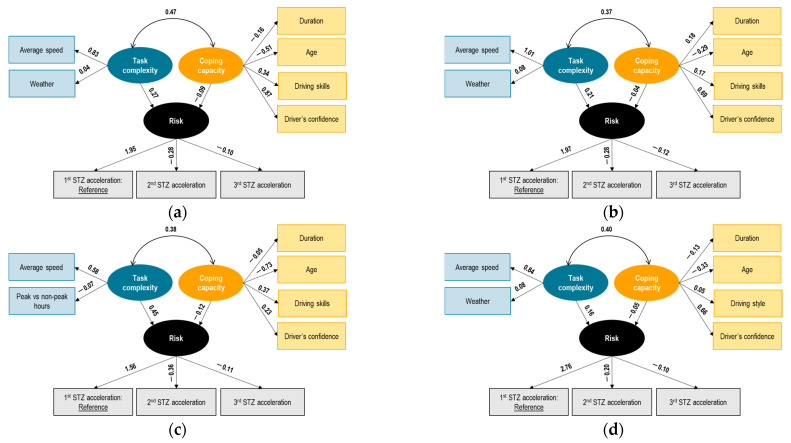
Results of SEM on risk—Belgian truck drivers—experiment phase 1 (**a**), 2 (**b**), 3 (**c**), 4 (**d**).

**Figure 5 sensors-23-09663-f005:**
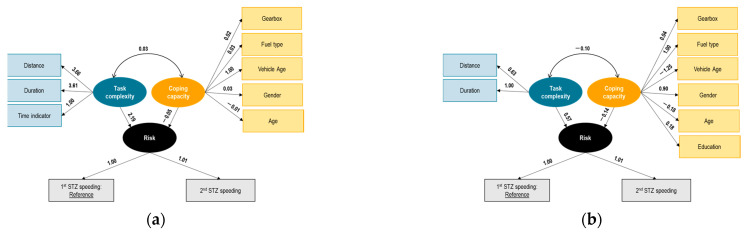
Results of SEM on risk—German car drivers—experiment phase 1 (**a**), 2 (**b**), 3 (**c**), 4 (**d**).

**Figure 6 sensors-23-09663-f006:**
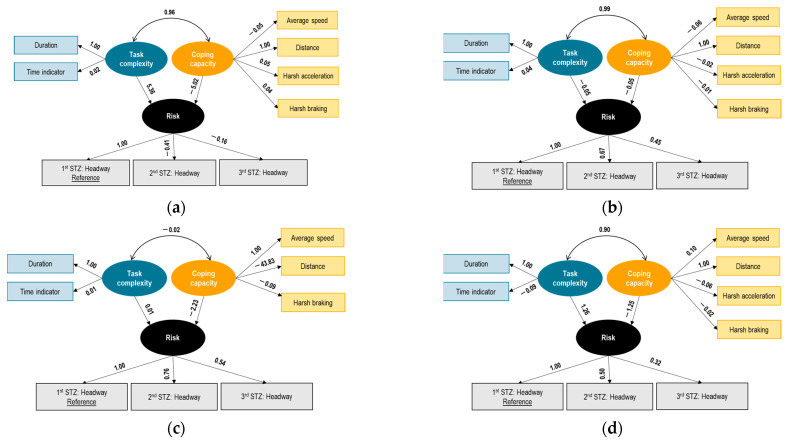
Results of SEM on risk—Portuguese bus drivers—experiment phase 1 (**a**), 2 (**b**), 3 (**c**), 4 (**d**).

**Table 1 sensors-23-09663-t001:** Parameter estimates and multicollinearity diagnostics of the GLM (Belgian trucks).

Variables	Estimate	Standard Error	z-Value	Pr(|z|)	VIF
(Intercept)	3.668	0.043	85.768	<0.001	-
Time indicator	0.908	0.078	11.683	<0.001	1.882
Weather	0.009	4.217 × 10^−4^	20.952	<0.001	1.228
High beam—Off	−0.018	7.062 × 10^−4^	−25.286	<0.001	1.470
Harsh acceleration	2.661	0.181	14.689	<0.001	1.013
Distance	−6.128 × 10^−4^	7.273 × 10^−5^	−8.426	<0.001	1.678
**Summary statistics**					
AIC	17,404.428				
BIC	17,413.817				
Degrees of freedom	88,377				

**Table 2 sensors-23-09663-t002:** Parameter estimates and multicollinearity diagnostics of the GLM (German cars).

Variables	Estimate	Standard Error	z-Value	Pr(|z|)	VIF
(Intercept)	1.105	0.057	19.549	<0.001	-
Duration	0.003	3.414 × 10^−5^	73.366	<0.001	1.262
Distance	5.735 × 10^−4^	3.723 × 10^−5^	15.404	<0.001	1.029
Harsh acceleration	1.282 × 10^−4^	1.974 × 10^−6^	64.951	<0.001	1.222
Fuel type—Petrol	0.219	0.010	21.446	<0.001	1.328
Vehicle Age	3.162 × 10^−4^	3.340 × 10^−6^	9.469	<0.001	1.277
Gender—Female	−0.275	0.021	−13.025	<0.001	1.256
Age	−0.003	0.001	−2.289	0.022	1.076
Drowsiness	1.009 × 10^−5^	2.656 × 10^−6^	3.800	<0.001	1.113
Time indicator	8.547 × 10^−5^	1.925 × 10^−6^	44.405	<0.001	1.080
High beam—On	0.817	0.059	13.963	<0.001	1.073
**Summary statistics**					
AIC	127,971.813				
BIC	127,981.881				
Degrees of freedom	174,299				

**Table 3 sensors-23-09663-t003:** Parameter estimates and multicollinearity diagnostics of the GLM (Portuguese buses).

Variables	Estimate	Standard Error	z-Value	Pr(|z|)	VIF
(Intercept)	3.441	0.020	168.858	<0.001	-
Time indicator	0.164	0.008	21.306	<0.001	1.002
Harsh braking	0.294	0.082	3.594	<0.001	1.051
Harsh acceleration	0.490	0.112	4.371	<0.001	1.052
Fatigue	−0.095	0.008	−12.527	<0.001	1.378
Distance	0.010	1.038 × 10^−4^	99.797	<0.001	1.379
**Summary statistics**					
AIC	153,657.374				
BIC	153,668.223				
Degrees of freedom	380,656				

**Table 4 sensors-23-09663-t004:** Model fit summary for different counties, transport modes, and experimental phases.

Model Fit Measures	Phase 1	Phase 2	Phase 3	Phase 4
Belgian (Trucks)				
**AIC**	2730.212	6417.821	3177.783	6089.699
**BIC**	2730.234	6417.839	3177.802	6089.713
**CFI**	0.921	0.813	0.882	0.843
**TLI**	0.881	0.719	0.778	0.764
**RMSEA**	0.062	0.088	0.064	0.077
**Hoelter’s critical N (α =0.05)**	386	197	372	256
**Hoelter’s critical N (α = 0.01)**	456	232	439	302
**German (Cars)**				
**AIC**	813,827.574	676,463.527	282,420.347	525,983.888
**BIC**	814,118.257	676,746.197	282,625.175	526,243.996
**CFI**	0.981	0.960	0.996	0.978
**TLI**	0.974	0.944	0.993	0.966
**RMSEA**	0.079	0.117	0.059	0.100
**Hoelter’s critical N (α = 0.05)**	234.136	106.728	507.651	153.470
**Hoelter’s critical N (α = 0.01)**	270.935	123.417	637.688	180.957
**Portugal (Buses)**				
**AIC**	3.328 × 10^6^	1.699 × 10^6^	1.511 × 10^6^	1.594 × 10^6^
**BIC**	3.328 × 10^6^	1.699 × 10^6^	1.511 × 10^6^	1.595 × 10^6^
**CFI**	0.983	0.985	0.998	0.964
**TLI**	0.974	0.978	0.997	0.946
**RMSEA**	0.053	0.052	0.019	0.051
**Hoelter’s critical N (α = 0.05)**	533.123	556.489	4284.444	582.268
**Hoelter’s critical N (α = 0.01)**	629.053	656.631	5188.355	687.057

## Data Availability

The data can be provided upon request.

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
