# Peer review of "Examination of the Effect of Task Complexity and Coping Capacity on Driving Risk: A Cross-Country and Transportation Mode Comparative Study"

_sensors, 2023, doi:10.3390/s23249663_

Round 1
Reviewer 1 Report
Comments and Suggestions for Authors
Dear Editor,
I would like to thank the authors of the manuscript ID No sensors-2670027. entitled “Examination of the effect of task complexity and coping capacity on driving risk: A cross-country and transportation mode comparative study” for presenting the results of their study on modeling the inter-relationship among driving task complexity, operator and vehicle coping capacity, and crash risk.
The manuscript presents the results of a naturalistic driving study based on data from 80 drivers of different vehicle types (trucks, cars, buses) from three countries (Belgium, Portugal, Germany), who were continuously monitored for the risk factors associated with task complexity (e.g., traffic conditions and weather) and coping capacity (e.g., driver's mental state, driving behavior, and vehicle status) through approximately 19.000 road trips. Driving behavior and the influence of real-time interventions (i.e., in-vehicle warnings) and post-trip interventions (i.e., post-trip feedback and gamification) on driving behavior, were thoroughly investigated as part of the on-road I –DREAMS experiment. The authors used the integration of Structural Equation Modeling (SEM) and Generalized Linear Models (GLM) in order to explore associations between the variables utilized for defining task complexity and coping capacity (vehicle and operator state), as well as the variables employed to represent driving risk.
After reading this article in detail, my main impression is that the article is well written, in adherence to the Journal’s standards
This study showed that the coping capacity of drivers increased with the increase in task complexity levels. Drivers tend to effectively manage their ability to anticipate and handle potential challenges when faced with demanding conditions while driving. Greater task complexity corresponded to increased risk. Conversely, coping capacity and inverse risk displayed a negative relationship across all phases, indicating that higher coping capacity was associated with reduced risk. Overall, the interventions positively impacted risk reduction by enhancing the coping capacity of the drivers.
There are several issues for consideration:
1. Title: Adequate. No remarks
2. Abstract: Adequate. No remarks
3. Introduction: Adequate. No remarks
4. Literature review: Adequate. No remarks
5. Data description: Adequate. Remarks:
a. No demographic data on drivers, all professional or car drivers were amateur drivers?
b. When describing variables, authors focused only on task complexity, with no details on coping capacity – vehicle or operator state variables? What is the reason for that?
c. Coding of the most important variables should be part of the data description, instead of randomly injecting explanations in the Results section
6. Methods: Too long and inconclusive, almost 4 pages of explanations of GLM, SEMs, and goodness of fit measures, instead of focusing on how the authors used these methods in this concrete study.
The usefulness of GLM and SEMs was already discussed in the literature review part of the manuscript.
7. Results: Not adequate. Remarks
a. The authors presented the results mixed with methodology (explanation and coding of variables: time indicator, wipers, high beam usage, etc) and discussion (giving possible explanations and comments) making it hard to follow the Results themselves.
b. It is not clear which phase of the study was analyzed in GLMs, baseline, or all four phases together?
c. Some of the comments of the Results were contradictory to real-life situations, such as drivers speeding more often in rainy weather, or low visibility conditions. Or the comment that the longer a person drives, the more likely they are to experience fatigue, leading them to drive at reduced speeds and exercise greater caution. Fatigue is a known factor for speeding and lower focus on driving.
d. What is an intercept variable?
e. SEM analysis –a lot of coding, repetition (Risk is measured by means of the STZ levels for headway (level 1 ‘normal driving’ used as the reference case; level 2 refers to ‘dangerous driving’, while level 3 refers to ‘avoidable accident driving’, The latent variable task complexity is measured by means of the environmental indicator of “ME_AWS_time_indicator_median” (indicating the time of the day) and total duration. It should be noted that based on the definition of task complexity, road layout, time, location, traffic volumes, and weather variables should be included in the analysis. However, road type (i.e. urban, rural, highway), location, traffic volumes (i.e. high, medium, low), and weather were not available …), discussion of the results that make it hard for the reader to follow just the Results stream.
8. Discussion: Not adequate. Remarks:
a. Short, no references or comparisons to other studies results
9. Conclusions: Not adequate. Remarks:
a. Three passages of rewriting the results and one passage from/for the Introduction
10. References: Not adequate, relevant. Out of 25 references, 11 are general statistics references
- Tables and Figures: Adequate.
Reviewer 2 Report
Comments and Suggestions for Authors
The i-DREAMS project established a 'Safety Tolerance Zone (STZ)' to maintain operators within safe boundaries through real-time and post-trip interventions, based on the crucial role of the human element in driving behavior. It has certain research significance and value.
A naturalistic driving experiment was carried out involving 80 drivers from Belgium, Germany, and Portugal and a large database of 19,000 trips and 847,711 minutes was created,with a huge workload.
However, the following problems exist:
(1) When collecting data and information from three countries, can we collect driving behavior related to a variety of models, rather than only collecting driving data of a single model in one country.
(2) Whether the generalized linear model or structural equation model can be improved, with emphasis on innovation.
The overall structure of the paper is reasonable and clear, and the workload is large, which has certain research significance and practical value.
Comments on the Quality of English LanguageMeet the requirements of journal English Writing
Reviewer 3 Report
Comments and Suggestions for Authors
The work is devoted to discussing a research grant on the influence of various parameters on drivers from several countries.
The authors have previously collected a large dataset and now they study the influence of various parameters on the result.
It’s not very clear how all this relates to the theme of the Sensors journal; for example, I would be more interested in knowing exactly how the data was collected.
To assess the influence of parameters on the result, the authors use fairly simple linear models and there are few parameters there; this could, of course, be enough to set the authors’ task, but at the current time, models that take into account hundreds of parameters and nonlinear relationships between them would be much more interesting. For such an analysis one already needs ML and working with big data, which would be relevant, but this is not in the article.
Further, the formulas are not made in the form of formulas, but in the form of text and look bad.
The drawings are no good, they are low resolution and look very bad on a high dpi screen (I have a Mac).
The analysis of related works does not take into account either data mining or modern big data analysis methods, and the reference list is clearly incomplete.
I have seen similar works before, and they also have not been analyzed in any way.
The dataset is not provided by the authors, although now everything in the world is moving towards open science.
Apart from stating the facts, these results are not very clear how to use them.
Reviewer 4 Report
Comments and Suggestions for Authors
Dear Authors,
the road safety is a very important issues. Studies from 3 different countries show interesting results that are important to many researchers in this area. Despite the authors' great care, the work requires several improvements.
In such a scientific work on such a widely discussed research area as transport safety, it is worth supplementing the literature with additional references from other Central European countries.
In the field of road safety, you can supplement your work with the following references:
https://doi.org/10.26552/com.C.2019.1.68-73
10.1109/AUTOMOTIVESAFETY47494.2020.9293531
https://doi.org/10.20858/sjsutst.2021.110.4
https://doi.org/10.14669/AM.VOL87.ART4
You mention the great potential for autonomous vehicle transport safety, interesting research presents, among others:
https://doi.org/10.3390/en14185778
https://doi.org/10.3390/ijerph20054559
They can be supplemented in lines 48-56.
We are used to a simple division of content into: Introduction, Methodology, Results and Discussion, Conclusion. The manuscript has a much more extensive division, but I have doubts whether the results chapter should not be together with the discussion chapter. The discussion is also not carried out, in fact you don't compare any other results with yours, why? This section is very important. Suggested references may also be helpful.
The conclusions section is very long. Maybe it is worth considering moving some issues to the discussion section?
Please add more references. Also, out of 25 references, 11 are general statistics references, add more research articles.
Thank you!
Author Response
Please see the attachement

Round 2
Reviewer 3 Report
Comments and Suggestions for Authors
The authors provided a long explanation in which they justified why they would not make the changes suggested by the reviewer.
I'm okay with papers describing consistent results, but that's not enough to be submitted to a good journal.
In principle, I am ready to agree with this, provided that:
1) All equipment (including all sensors), methods and algorithms for data collection should be described explicitly in a separate section with schemes - and not as a superficial description has now been added!
2) Examples of anonymized datasets must be provided and opened, and using their example, their derivatives must be calculated by authors' method in a special section of the data description.
3) All!! the pictures should be redone, they are of low quality with jpeg artefacts, please use vector graphics and export with a good resolution of 300-600dpi, not 72dpi. Look, I have attached an example of enlarging your picture in .pdf, so the font in the picture should be of the same quality as the text font next to it.

Reviewer 4 Report
Comments and Suggestions for Authors
Dear Authors,
thank you for your corrections to the manuscript. Nowadays, the work is much better and more readable. The issues included in the review have been clarified and supplemented.
Thank you!
Author Response
We would like to thank the Reviewer for the valuable recommendations made for the present study!